# Improving the Indoor Air Quality in Nursery Buildings in United Arab Emirates

**DOI:** 10.3390/ijerph182212091

**Published:** 2021-11-18

**Authors:** Mohammad Arar, Chuloh Jung

**Affiliations:** Department of Architecture, College of Architecture, Art and Design, Ajman University, Ajman 346, United Arab Emirates; m.arar@ajman.ac.ae

**Keywords:** Indoor Air Quality (IAQ), nursery, air purifier, finishing material change, United Arab Emirates

## Abstract

Children inhale indoor air at 400 mL/min∙kg per body weight, 2.76 times more than adults. They have weaker immunity than adults and are more exposed to asthma, allergies, and atopic diseases. The objective of this paper is to suggest effective management and improvement measures for indoor air quality for nurseries. As a methodology, 16 nurseries (total of 35 classrooms) were selected to measure the indoor air quality compared with WHO IAQ Standard, and identify the daily concentration change of the pollutants. Based on the measurements, IAQ improvements for selected facilities are carried out to compare the results before and after improvement. The result has shown that the concentration of Carbon Dioxide (CO_2_), Total Volatile Organic Compounds (TVOC), Total Suspended Particles (TSP) and formaldehyde (CH_2_O) exceeds WHO IAQ standards. The concentration of CO_2_ and TSP is changed mainly by physical activity of children and that of CH_2_O and TVOC is changed mainly by ventilation after school start. TVOC decreased by 46.4% and the TSP decreased by 21.7% after air purifier, but CH_2_O and TVOC increased 1.8–3.8 times after interior renovation with low-emission finishing materials. After new ventilation installation, the CH_2_O and TVOC reduced half and the TSP reduced one third. It is proven that the most effective way to reduce the concentration of air pollutants in nurseries is the installation of a new ventilation system, followed by an air purifier. The renovation with low-emission finishing materials cannot improve IAQ in a short period of time.

## 1. Introduction

It is essential to manage Indoor Air Quality (IAQ) in buildings to create a pleasant environment and maintain the health of occupants [1,2,3]. Unlike office buildings, where most of the occupants are adults, it is important to maintain pleasant indoor air quality in the nursery, where most of the occupants are children and play in groups [4,5,6]. Infants and children inhale indoor air at 400 mL/min∙kg per body weight, compared to 150 mL/min∙kg per body weight of adults [7,8]. They are highly influenced by the indoor environment and have weaker immunity than adults, so they are more likely to suffer from asthma, allergies, and atopic diseases that can be caused by indoor air quality [9,10]. Atopic asthma and wheezing in infants and children, which are judged to be environmental diseases, can affect lung growth, and may impair lung function [11,12,13]. As allergic diseases can develop into more serious types of allergic diseases as they progress naturally, it is very essential to manage indoor air quality in childcare facilities [14,15,16].

The Public Health and Safety Department of Dubai Municipality assessed the Indoor Air Quality (IAQ) for 70 educational institutions, universities, schools, nurseries, kindergartens, and health care centers between 2013 and 2014 [17,18,19]. Based on this assessment, Dubai municipality established the IAQ (Indoor Air Quality) stipulation with less than 0.08 ppm (parts per million) of formaldehyde (CH_2_O), less than 300 micrograms/m^3^ of TVOC (Total Volatile Organic Compound), and less than 150 micrograms/m^3^ of suspended particulates (less than 10 microns) in 8 h of continuous monitoring prior to occupancy [20,21]. Even though a strict IAQ standard has been set for childcare facilities, they are not adhered to in practice [22,23]. Exceeding these IAQ standards can not only lead to decreased learning efficiency in infants and children, but also serious SBS (Sick Building Syndrome) symptoms such as respiratory diseases, nausea, eye irritation, drowsiness, and long-term health problems [24,25]. To reduce the adverse effects on the health of infants and children due to Indoor Air Quality (IAQ) pollution, three methods to keep indoor air quality clean in nurseries are used such as controlling the source of pollutants, improving ventilation, and removing pollutants via purification [26,27,28]. Many studies have been conducted for these three general methods [29,30]. However, most of the research targets houses and offices, and there is very little research on indoor air quality in childcare facilities, used by infants and children with weak immunity [31,32,33]. In addition, while research has been focused on improving indoor air quality before moving in, studies conducted for occupied space are very rare [34,35].

This study aims to determine the most effective methods to enhance the Indoor Air Quality (IAQ) of the nurseries among placing an air purifier, changing finishing materials (low-emission materials), and installing ventilation facilities. IAQ was measured on site for 16 facilities to define the current status of IAQ in nurseries in the UAE. Six nurseries were measured during a specific time period to understand the daily concentration change for indoor air pollutants. Sixteen nurseries had installed three different IAQ improvement methods (placement of air purifiers (10), change of finishing materials (3) and installation of ventilation facilities (3)) according to the measurement data, characteristics of nurseries, and manager’ s requirements. After improvement, the indoor air quality was re-measured to identify how these three methods effectively reduce indoor air pollutants in nurseries for future IAQ management (Figure 1).

## 2. Materials and Methods

The fundamental methods to solve the indoor air pollution problem include control of the source, improvement of ventilation, and control by purification [36,37,38]. Examples of source control include the use of low-emission building finishes and bake-outs [39,40]. Ventilation improvement uses natural ventilation and mechanical ventilation to increase the indoor inflow of outdoor air with low pollution level [41,42]. Alternatively, if the outdoor air pollution is serious, there is a method to purify the indoor and outdoor air, and to introduce and recirculate it indoors [43]. Removal control is a method of adsorbing and filtering pollutants or decomposing them by catalysts [44]. There is a method to improve indoor air quality on its own using air purifiers, catalyst materials and air purifying plants [45].

As a study to improve indoor air quality, there is a study result in which the concentration of TVOC and CH_2_O was reduced by more than 20% compared to the initial concentration after bake-out in a new apartment building in relation to source control [46,47]. A study was conducted on the effect of bakeout of general finishing materials and low-emission finishing materials [48]. To increase the effect of reducing pollutants by bake-out, the main influencing factors and operating conditions were reviewed [49].

As a study on ventilation improvement, after identifying the emission intensity of pollutants (radon) from building materials for residential buildings, the radon concentration according to the location of the room and the opening and closing state of the opening was identified through simulation, and ventilation was performed to improve indoor air quality [50]. In addition, there is a study result that the location of the seal and the radiation time of contaminants should be considered [51]. In the construction stage before moving into an apartment, natural ventilation, mechanical (exhaust) ventilation, and natural ventilation and mechanical (exhaust) ventilation were divided into ventilation methods, and the indoor air quality was improved before ventilation and 3 weeks after each ventilation method was applied [52,53]. There is a study comparing the reduction efficiency according to each ventilation method by measuring it [54]. Additionally, by measuring and simulating the ventilation rate and CO_2_ generation in the classroom, there is a study result that CO_2_ concentration decreases when the summer temperature is set low, and the CO_2_ concentration increases when the winter temperature is set high [55,56].

Previous research has conducted ventilation methods such as natural ventilation, air conditioner operation, mechanical ventilation, and measure indoor air quality, and concludes natural ventilation dilutes pollutants under the influence of outside air [57]. Although air conditioners and mechanical ventilation have lower ventilation rates, studies have reported that pollutants are lowered in the process of air recirculation, thereby improving indoor air quality [58].

Studies on the improvement effect of purification have been actively conducted on the reduction of indoor air pollutants by air purification plants [59,60]. For example, there was a study result that proved that the CH_2_O concentration was reduced when air purifying plants were planted indoors by conducting a closed experiment assuming outdoor air conditions in which natural ventilation is difficult in winter and early spring [61,62]. In addition, there is a study comparing the concentration of pollutants due to the decomposition of pollutants in indoor ventilation and photo plasma, and there is a study that the removal of pollutants is more effective than when the ventilation is performed twice and 1.3 times [63]. It has been reported that when the photo plasma device is operated, pollutants are greatly reduced compared to non-operated threads, and pollutants can be reduced in a short time [64]. It has been investigated that ventilation lowers the concentration of all types of VOCs, but photocatalysis lowers the concentration of only a few VOCs [65,66].

Concerning ventilation control, there is a study that evaluates the effect of reducing indoor air quality pollutants by selecting six two-bedroom households to understand the effect of natural ventilation and decomposition agent construction for a newly built apartment building before moving in [67,68]. In the household with natural ventilation, the rate of increase in pollutants was reduced by 2/3 compared to the sealed household [69]. In addition, a study showing that the concentration of VOCs decreased in the houses using the ventilation system and cement floors instead of carpets by installing ventilation facilities and changing materials in each house targeting four houses requiring improvement of indoor air quality has been reported [70].

The improvement plans studied above are mainly those that can be implemented during the construction phase and before moving in [71]. There were proposals that had to be implemented by contractors rather than the improvements that users can generally implement [72]. Therefore, it is necessary to study the indoor air quality improvement plan, which can be implemented relatively easily by occupants in a building already in use, and the verification of the effect thereof [73].

In addition, most of the studies on indoor air quality improvement have mainly focused on residential buildings, but there are few studies on the nursery IAQ [74]. As a study on nurseries, there is a study that TVOC, CH_2_O, airborne bacteria and CO_2_ exceed the standard in nursery and elementary school buildings in Greece [75]. There was a study that measured temperature, humidity, CO_2_, CO, and PM_10_ for 10 childcare facilities, investigated the maintenance status, and identified factors affecting the concentration of each pollutant. However, it is considered that research is needed to improve the indoor air quality in the nursery, as it is mainly limited to the actual situation report. Therefore, a method to improve indoor air quality in nurseries where children stay for a long time is explored. Based on the basic indoor air quality improvement plan, the effects of installing air purifiers, changing building finishing materials, and installing ventilation facilities that managers can easily implement are investigated.

Sixteen nurseries were selected based on the EdArabia’ s best nurseries rankings and number of reviews on the site for Dubai (Figure 2), Sharjah (Figure 3) and Ajman [76]. Temperature, humidity, CO_2_, TSP, CH_2_O, and Volatile Organic Compounds (VOCs) were measured in a total of 35 classrooms. For each indoor air quality pollutant in the target room, the degree of contamination was compared with the standard value of the WHO IAQ Standard, and the daily concentration change of the pollutant was identified. Based on the measurement results and the manager’ s request, to improve the indoor air quality of the target facility, an air purifier or ventilation system was installed, or the finishing material of the facility was changed to an eco-friendly building certification material. After improvement, the degree of reduction in pollutant concentration and satisfaction with the improvement were investigated via air quality measurement.

Table 1 shows basic information such as the number of children and the year of establishment of the target facility, as well as the maintenance status of mechanical ventilation devices and natural ventilation times expected to be related to indoor air quality. 01-FU–10-HO are facilities that we improved indoor air quality, and the air quality was measured before and after the improvement. 11-OR–16-BL facilities are measured for the daily concentration change of pollutants. The target facilities are 10 facilities in Dubai, 4 facilities in Sharjah, and 2 facilities in Ajman. By area, there are two facilities under 500 m^2^, 10 facilities with 500–1000 m^2^, and 4 facilities with more than 1000 m^2^. In terms of the number of children, there are eight facilities with 50 or less children, six facilities with 50 to 100 children, and two facilities with 100 to 150 children. By year of completion, there are two facilities before 2005, 12 facilities between 2006 and 2015, and two facilities after 2016. Among the target facilities, 17 facilities have a mechanical ventilation system comparable to that of a local exhaust system in kitchens and toilets; more than half of them. None of the facilities had ventilation in the classroom. 8 out of 16 childcare facilities had air purifiers in each classroom. Measurements were made before and after improvement in 32 classrooms in a total of 16 childcare facilities. Hourly measurements were made in 10 classrooms in six facilities.

The measured indoor air pollutants were CO_2_, TVOC, CH_2_O, and TSP with temperature and humidity. The measurements were conducted in 32 classrooms in 16 nursery facilities; the most common two nursery rooms where children reside, and all measurement points were set at a height of 1.0 to 1.2 m from the floor in consideration of the child’ s breathing line. Indoor air quality was measured in two stages: before and after improvement of the indoor air environment. Facility 11-OR–16-BL between February and November 2020, Facility 01-FU–10-HO between February and November 2020, pre-measurement, improvement work and re-measurement after improvement. The measurements before improvement were conducted in the spring, February to March, and measurements after improvement were carried out in October to November, autumn, when the outdoor temperature was relatively the same to reduce seasonal effects. The measurements before and after improvement were conducted from 08:00 a.m. to 12:00 p.m. in the morning after children go to school. During the summer months of June and August, indoor air quality improvement work was carried out in nursery facilities.

In addition, to analyze the change in the concentration of pollutants according to time period, two classrooms in 6 facilities (11-OR–16-BL) were targeted in March 2020, the same time as the measurement before improvement. The measurement was conducted three times, before going to school (before 08:00 a.m.) in the morning (08:00 a.m. to 12:00 p.m.), and in the afternoon (12:00 p.m. to 16:00 p.m.) to avoid mealtimes and times when children are not present. To achieve the maximum objectivity of the experiment, there were no intervention or control of children’s activities for the measurement.

The method of measuring pollutants before and after the improvement and according to the time is the same. Temperature, humidity, and CO_2_ were measured 30 times for 1 min for 30 min by direct reading method. TSP was measured 10 times for 3 min each. TVOC and CH_2_O were measured according to the WHO IAQ standard test method. The test method is a method of measuring two or more points at 1 m from the inner wall and floor surface of a place judged to be representative of the pollution level of the target facility during the daytime (08:00 a.m. to 19:00 p.m.).

Table 2 shows the measuring devices and methods according to the measurement items. The collection of air during sampling was carried out continuously twice in one or two chambers for 30 min using Formaldehyde Meter HFX205-100. CH_2_O was collected through Formaldehyde Meter HFX205-100 at a flow rate of 500 mL/min and then analyzed using HPLC. TVOC was collected through VOC Environmental Meter PCE-VOC 1 at a flow rate of 100 mL/min and analyzed using GC/MS (Varian-SATURN2200/Shimadzu-QP2010). When measuring indoor air quality before and after improvement work, day-to-day childcare activities were carried out, so there was no control over changes in equipment used indoors, such as teaching aids and toys, and the amount of activity of occupants.

## 3. Results

When measuring indoor air quality before improvement, the average and standard deviation of the room temperature were 26.6 °C and 1.9, respectively, and the rooms were measured at relatively similar temperatures. The mean and standard deviation of humidity were 41% and 16.87, which showed a relatively large standard deviation compared to temperature.

### 3.1. Indoor Air Quality Measurement Results by Pollutants before Improvement

Figure 4, Figure 5, Figure 6 and Figure 7 show the concentrations of CO_2_, TSP, CH_2_O, and TVOC measured in each nursery room and outside air before improvement. When measured, the average outdoor air CO_2_ concentration was 436.7 ppm, indicating that the pollution level of the outdoor air was not serious. The CO_2_ concentration of seven facilities in 01-FU, 04-ID, 08-KI, 09-AK, 10-HO, and 14-RO and 11 nursery rooms (37%) exceeded 920 ppm, the CO_2_ standard of the WHO IAQ standard. The source of CO_2_ pollution is due to the breathing of infants and children in the room. Among them, in the case of facility 01-FU and facility 14-RO, both classrooms approached or exceeded 2000 ppm, twice the CO_2_ maintenance standard, indicating that the degree of CO_2_ contamination was serious (Figure 4).

The average TSP (PM_10_) of the outdoor air at the nurseries was 90.29 μg/m^3^, indicating that the overall level of contamination was lower than WHO IAQ standard. The measurement results of 08-KI (141.3 μg/m^3^) and 16-BL (166.7 μg/m^3^) facilities were very high. The surrounding areas of 08-KI and 16-BL are industrial and commercial areas, and it is understood that the outside air is polluted by the influence of dust from surrounding construction sites and factories, as well as a lot of vehicle flow. Room 1, Room 2 of 08-KI, and Room 2 of 16-BL were seriously polluted indoors with the concentration of TSP at 165.6 μg/m^3^, 182.0 μg/m^3^, and 184.1 μg/m^3^ due to the pollution of the outside air along with the dust generated indoors. Although the concentration of TSP in the outdoor air was high in 16-BL room 1, it was measured to be lower than the TSP concentration in the outdoor air by keeping the room clean. In the case of 14-RO room 2, the outdoor air TSP concentration was low, but the amount of dust generated indoors was large, so the TSP concentration was measured to be high as 129.1 μg/m^3^. Other 01-FU room 2, 02-DO room 1, 10-HO room 1, 11-OR room 1 and 12-AM room 2 exceeded the standard value of 100 μg/m^3^ (Figure 5).

The TVOC concentration in the outside air of the target facilities was 512.89 μg/m^3^ in the case of the 15-GA, indicating a high level of pollution. In case of the rest of the nurseries except for 15-GA, the average level of contamination was 125.06 μg/m^3^, indicating below WHO IAQ standard. It was measured that 8 out of 16 facilities (02-DO, 05-KA, 07-LL, 08-KI, 09-AK, 14-RO, 15-GA) and 12 out of 32 classrooms exceeded the recommended standards. The place with the highest level of TVOC pollution was found to be Room 1 and Room 2 of the 15-GA facility. Each concentration was 1583.39 μg/m^3^ and 1,473.70 μg/m^3^, which was more than three times the recommended standard of 400 μg/m^3^ (Figure 6).

The CH_2_O concentrations in the outside air of 02-DO, 07-LL, and 12-AM were 99.64 μg/m^3^, 132.8 μg/m^3^, and 234.2 μg/m^3^, respectively, and the degree of contamination was serious compared to other nurseries. As a result, the indoor CH_2_O concentration was also measured to be high. 07-LL, 11-OR, 12-AM, and 15-GA were found in places that exceeded the WHO IAQ standard of 100 μg/m^3^. In the case of 11-OR and 15-GA, the indoor CH_2_O concentration is higher than the outdoor air, so it is judged that they are contaminated by various finishing materials and CH_2_O emitted from teaching materials (Figure 7).

### 3.2. Changes in Indoor Air Pollutants Concentration over Time

Figure 8, Figure 9, Figure 10 and Figure 11 show the changes in the concentrations of pollutants CO_2_, TSP, CH_2_O, and TVOC measured over time in each of the two classrooms at six nurseries. There was a difference in the degree of contamination for each material in each nursery room, but the change pattern of the concentration according to time was similar.

In case of CO_2_, before school (798.3 ppm), in the morning after school start (1556 ppm), and in the afternoon during school (1197.5 ppm) was measured according to time. As for the change trend of the CO_2_ concentration in each nursery room, as shown in the average concentration change pattern, the concentration was the lowest before school, and the concentration increased after school start, and then decreased again in the afternoon (Figure 8). As for the daily change in TSP concentration, such as the trend of change in CO_2_ concentration, the concentration of TSP before school was the lowest, and it rose the highest in the morning after school, and then decreased in the afternoon. However, the concentration continued to increase over time in 4 out of 12 nursery rooms. The hourly average concentrations were measured to be 47.2 μg/m^3^, 85.3 μg/m^3^, and 65.0 μg/m^3^, respectively (Figure 9).

The concentration changes of CH_2_O with time gradually decreased with time at all measurement points, except for one room. The average concentration by time was 108.6 μg/m^3^ before school, 96.6 μg/m^3^ in the morning after school start, and 49.4 μg/m^3^ in the afternoon during school. In the classroom with the most severe indoor air pollution by CH_2_O, the concentration changes were 363.5 μg/m^3^, 249.5 μg/m^3^, and 147.8 μg/m^3^, respectively (Figure 10). TVOC concentrations, like CH_2_O, gradually decreased over time. The concentration changes with time were 108.6 μg/m^3^, 96.6 μg/m^3^, and 49.4 μg/m^3^, respectively, on average. The room with the highest TVOC measurement was the same as the facility with the highest CH_2_O, and the measured values changed to 3364.86 μg/m^3^, 1268.15 μg/m^3^, and 1109.31 μg/m^3^ (Figure 11).

### 3.3. Improvements for Each Nurseries

The improvement plan for indoor air quality carried out in this paper are (1) “installation of an air purifier” to remove indoor air pollutants, (2) “change of building materials” to remove pollutants (replacement with finishing materials that emit less pollutants), and (3) “installation of ventilation equipment” to dilute the polluted air. The improvement work carried out at each nursery was determined in consideration of the indoor air quality measurement results, the manager’ s opinion, and the situation of the nurseries. Table 3 below shows the facilities according to the indoor air quality improvement working group.

Group A consisted of a total of 10 nurseries (01-FU–10-HO), which do not have air purifiers. The air purifier is the most preferred improvement plan by facility managers because of the ease of installation. The device was adapted to the size of the nursery so that the children could use it continuously while they were in the room. The installed air purifier has a built-in deodorizing filter that removes odors and indoor air pollutants and an anti-virus HEPA filter that can filter out more than 99.9% of 0.3 μg fine particles.

Group B, which is a change of finishing materials, targeted 11-OR and 12-AM nurseries with CH_2_O concentrations higher than the standard, and 13-DR nursery that were very old and poorly managed. As for the replaced eco-friendly building materials, the best certified materials were used with the emission of CH_2_O and TVOC less than 0.015 mg/m^2^ h and 0.1 mg/m^2^ h, respectively, through the chamber test. In facilities 11-OR and 13-DR, wallpapers were replaced on the four sides of the nursery room walls, and in 12-AM, the flooring materials were changed.

In Group C, which is expected to be most effective in improving indoor air quality, new ventilation equipment was installed since two or more of the measured pollutants exceeded WHO IAQ standard, and the level of pollution was severe. In group C, the outdoor air was measured to be clean, but the 14-RO with high TSP and TVOC concentrations were measured in the nursery room, and the 15-GA and 16-BL with high outdoor air pollutant concentrations and two or more pollutants exceeding the WHO IAQ standard in the nursery room. In addition, at the time of the visit, it was confirmed that the nursery room had a closed floor plan, and the windows were very narrow. The ventilation system is a method of supplying and exhausting 250 CMH (Cubic Meter per Hour) of fresh air into the room through a diffuser capable of temperature/humidity control and a filter. One device was installed in each nursery room so that natural and mechanical ventilation were performed simultaneously.

### 3.4. Comparison of IAQ According to Improvement Methods

To understand the degree of improvement in indoor air quality according to the improvement work in Group A–Group C, the concentrations before and after improvement were compared for each nursery. The degree of improvement was expressed as a percentage before and after the improvement using the following equation.
(1)$$Reduction\;Rate\;\left(RR\right)=\frac{m_{b}-m_{a}}{m_{b}}\times 100$$

*m_a_*: Measurement Results after Improvement*m_b_*: Measurement Results before Improvement

#### 3.4.1. Nurseries with the Installation of Air Purifier

Table 4 below shows the changes in indoor air pollutants before and after the improvement with air purifiers. The average reduction rate of pollutant concentration in nursery rooms with air purifiers was the highest in TVOC at 46.31% compared to before improvement. Next, TSP was 21.7%, CO_2_ was 18.16%, and CH_2_O was 13.7%. On average, the concentration of all contaminants decreased after improvement. However, CO_2_ and TVOC concentrations increased in three out of 16 nursery rooms, and TSP was measured slightly higher than before improvement in two rooms. There was a total of seven facilities with increased CH_2_O concentration, and it was the indoor air pollutant which had the highest concentration among all the pollutants. The concentrations of CH_2_O and TVOC in 09-AK were 84.9% and 86.4% in Room 1, and 91.0% and 96.7% in Room 2, respectively, showing the highest concentration reduction rates. CO_2_ concentration was reduced by 60.9% in Room 1 and 64.2% in Room 2. The main source of CO_2_ is human respiration or combustion. It is judged that the concentration is lowered not because of an air purifier but the decrease in the number of occupants and the amount of activity. In the case of TSP, the Room1 in 08-KI showed the highest reduction rate of 72.1%. The effect of improving indoor air quality by air purifiers was found to be the most effective in reducing TVOC. In case of CH_2_O, the concentration was reduced on average and the degree was found to be insignificant.

#### 3.4.2. Nurseries with Building Material Change

Table 5 shows the changes in pollutants before and after the improvement of the facilities with building material change. In case of using a finishing material certified with a low emission of pollutants, the concentrations of CH_2_O and TVOC after improvement were significantly lower than before improvement in all six nurseries. In the 11-OR-1 room, the decrease rate of CH_2_O concentration was −498.3%, which was about 5 times lower than before the improvement, and the decrease rate of TVOC concentration in L2 was −812.8%, which was about eight times lower. The average reduction rates of CH_2_O and TVOC were −178.6% and −380.7%, respectively. Considering that re-measurement was carried out about two months after changing the finishing materials at each nursery and that the amount of indoor air pollutants emitted by building materials was very high at the beginning and then decreased over time, The improvement effect regarding building material change in short period of time was difficult to expect based on many previous studies. If change of building material is considered, it should be changed in long break time such as summer vacation [77].

In addition, the 13-DR changed the finishing material at the request of the manager due to the deterioration of the material, so that the interior was aesthetically pleasing, but the indoor air quality became more serious compared to the previous one. The concentration of TSP in room 13-DR-2 decreased slightly after improvement to −5.6%, but the concentration of TSP in other nursery rooms decreased overall, resulting in an average decrease rate of 15.2%. This is the effect of cleaning the nursery room, furniture, and various items carried out along with the material change. The reduction rate of pollutants before and after CO_2_ improvement was −25.6% on average, and the CO_2_ concentration decreased about two times to −221.4% in 12-AM-2 room. Since the source of CO_2_ pollution is outside air or human breathing, it cannot be said to be an effect of material change.

#### 3.4.3. Nurseries with New Ventilation Installation

Table 6 shows the concentration of pollutants before and after the installation of ventilation equipment in the nursery and their reduction rate. In case of CH_2_O, the 15-GA-2 room showed the highest reduction rate of 79.7%. The lowest measured case was 17.46% in the 16-BL-1 room, and the average reduction rate was 53.5%, so the concentration was reduced most effectively among pollutants. Regarding CH_2_O, compared to other improvement methods such as air purifier placement and building material change, air purifier has the highest efficiency (91.06%) at 09-AK-2. The TVOC concentration decreased the second most with an average decrease rate of 52.94% before and after improvement. Compared to other improvement methods, air purifier has the highest efficiency (96.68%) at 09-AK-2. The concentrations of TSP and CH_2_O were reduced in all the measured rooms, and in case of TSP, compared to other improvement methods, new ventilation system has the highest efficiency (72.27%) at 14-RO-1. In case of CO_2_, the concentration reduction rate was relatively low, and the concentration increased after improvement in three of six rooms, indicating that the reduction effect by installing ventilation facilities was the least. Compared to other improvement methods, new ventilation system has the highest efficiency (75.2%) at 15-GA-2.

## 4. Discussion

There are various sources of indoor air pollutants and the improvement methods for various pollutants are also different. It is necessary to identify the problematic pollutants of nurseries, and to establish appropriate improvement measures. In addition, in order to understand the effect of a low-emission finishing material for future study, a study on the concentration change according to each elapsed time will be conducted when changing to a general material and a low-emission finishing material.

Facing the COVID-19 pandemic, IAQ became a more important social phenomenon [78]. After a new airborne virus ignited a global pandemic, IAQ and its impact on our health became one of the most crucial issues, even though the spotlight was primarily on the threat of COVID-19 pathogens [79]. In last two years, many global building certifications have focused on monitoring IAQ, especially on pathogens and particulate matter (PM) [80]. Since we are moving past the COVID-19 pandemic, it will be important to maintain higher standards, addressed during the pandemic. Regarding COVID-19 risk mitigation strategies, it is absolutely important not only to wear a mask, social distance, and outdoor de-densify, but also to have appropriate IAQ management systems such as increasing natural and mechanical ventilation, increasing filtration, managing indoor temperature and humidity, and frequent UV cleaning for HVAC units [81].

Our methodology to enhance the IAQ in 16 nurseries are the same as above and the result data can possibly be implemented to augment Dubai Municipality IAQ stipulation, not only for nursery building IAQ enhancement, but COVID-19 risk mitigation.

## 5. Conclusions

The nature of IAQ research is multidisciplinary with four main categories such as medicine, energy, buildings, and environments. The focus originated from a medical point of view such as the symptoms of SBS in places of work and recently shifted to the characterization of pollutants and risk assessment due to global trends of smoking bans in public spaces. Moreover, the focus was moved from office buildings to schools and hospitals. Currently there have been many attempts to use four IR (Industrial Revolution) technologies such as low-cost sensors to measure indoor air pollutants and the apply new technology to passively improve air quality. Since many unprecedented chemicals and new building material are used in daily life, various new monitoring campaigns and sampling methods have appeared.

This study aimed to measure indoor air quality in nurseries to determine the current situation, conduct indoor air quality improvement work for each facility, and compare the results before and after improvement by re-measurement of indoor air quality. The objective is to suggest effective management and improvement measures for indoor air quality for nurseries.

First, the concentration of CO_2_ and TVOC exceeding WHO IAQ standards in the target nurseries was 37% each, and the concentration of TSP and CH_2_O was 23%, respectively. Some nurseries were found to have very serious contamination levels for certain substances. In most nursery rooms, the indoor pollutant concentration was higher than the outdoor air, and it was found that the cause of indoor air pollution was greater indoors than outdoor factors. In addition, to manage and improve indoor air quality in nurseries, natural ventilation, or ventilation equipment to bring outdoor air to dilute the indoor pollutants is effective since the concentration of pollutants in the outdoor air is not high.

Second, the daily change in the concentration of pollutants increased in the case of CO_2_ and PM_10_ after children went to school, and the concentration decreased in the afternoon during a nap in the afternoon and children went home. The concentration of CO_2_ and TSP was changed mainly by the presence and physical activity of children. On the other hand, the concentrations of CH_2_O and TVOC were the highest before school start, and the concentrations decreased gradually after school finish. The concentration of pollutants was reduced by ventilation after the opening of the nurseries, and contaminants were emitted and accumulated in the closed classroom, such as building materials, furniture, and various play equipment. Therefore, to keep the indoor air quality clean in nurseries, sufficient natural ventilation should be performed before children attend.

Third, as a result of the air quality measurement before and after improvement, the TVOC concentration decreased significantly to 46.4% and the TSP concentration decreased by 21.7% in the nursery room with air purifier installation. This is due to a deodorizing filter that adsorbs and removes the pollutants and a HEPA filter which removes particulate pollutants. Concentrations of CH_2_O and TVOC increased by 178.6% and 380.7%, respectively, in nursery rooms where building materials were changed with low-emission finishing materials. Even for low-emission finishing materials, the initial emission of pollutants is higher than that of general materials already in use, so it is not effective to improve indoor air quality in a short period of time. In the nursery room whose materials were changed due to deterioration of the facility, indoor air quality pollution grew worse. In addition, the TSP concentration was also reduced by the cleaning work accompanying the material change. In case of nurseries with ventilation facilities, the CH_2_O concentration showed the highest reduction rate of 53.5% after improvement. The TVOC and TSP concentrations were reduced by 52.9% and 36.7%, respectively, indicating that the indoor air quality improvement effect was superior to that of other improvement plans. Additionally, the CO_2_ concentration was reduced the most by the ventilation system, but it was found to be relatively difficult to control, as it was largely affected by the number of occupants and the amount of activity in the nursery room.

As a way to improve indoor air quality in nurseries, the most effective way to reduce the concentration of pollutants was the installation of a ventilation system, followed by an air purifier. Since air purifiers will have different pollutant reduction rates due to the characteristics of the product, a device suitable for the characteristics of the facility should be used. The use of low-emission finishing materials cannot be a direct way to improve indoor air quality, but it is considered to be better than the degree to which indoor air quality is polluted by changing to general building finishing materials in nurseries.

## Figures and Tables

**Figure 1 ijerph-18-12091-f001:**
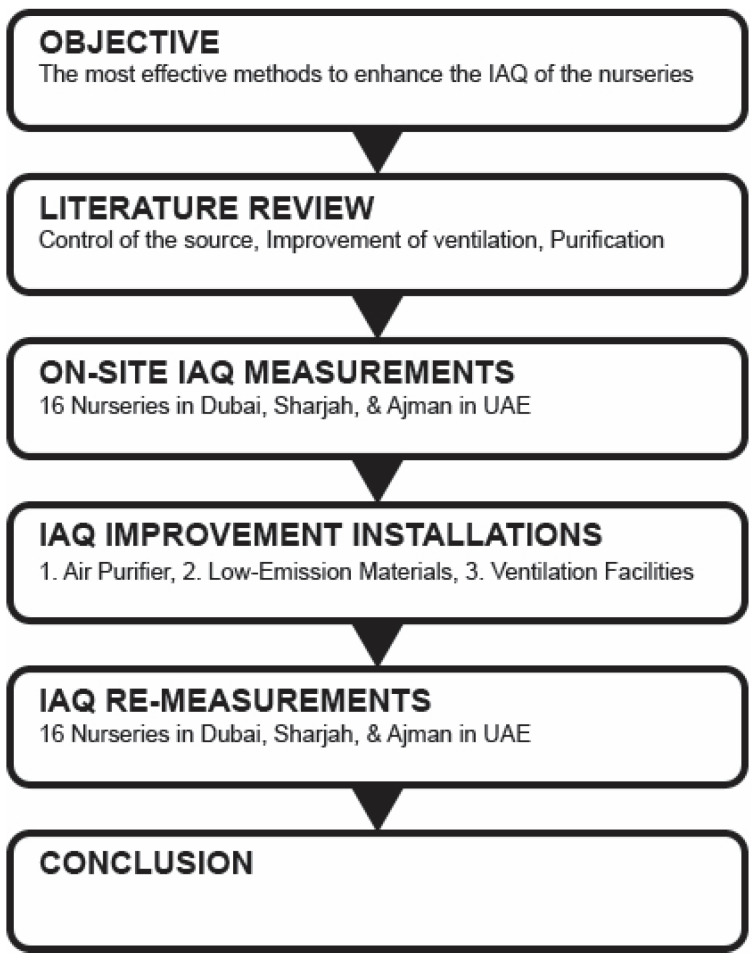
Research Process.

**Figure 2 ijerph-18-12091-f002:**
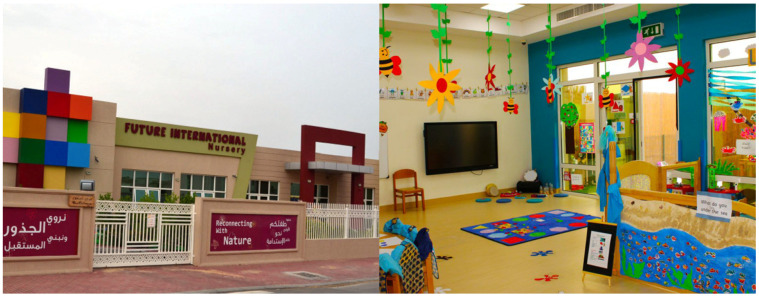
01-FU Nursery Exterior and Interior View.

**Figure 3 ijerph-18-12091-f003:**
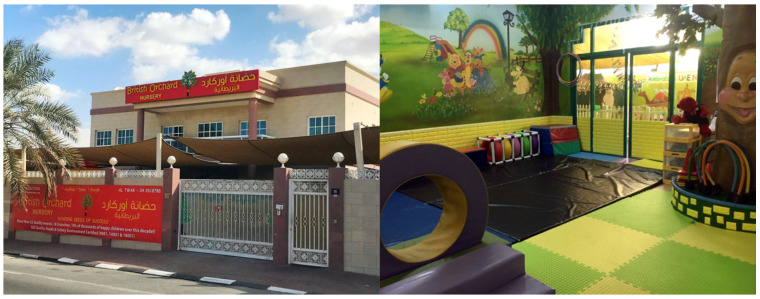
11-OR Nursery Exterior and Interior View.

**Figure 4 ijerph-18-12091-f004:**
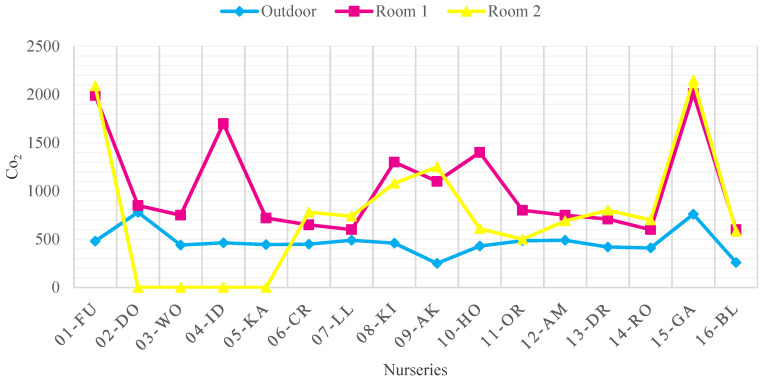
CO_2_ concentration in each room and outside air before improvement (WHO Standard: 920 ppm).

**Figure 5 ijerph-18-12091-f005:**
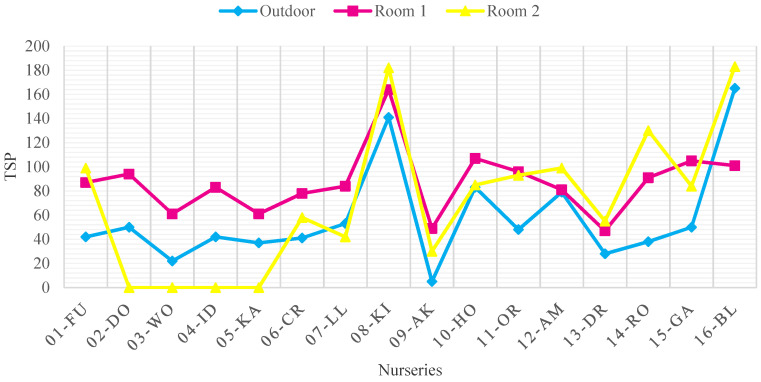
TSP concentration in each room and outside air before improvement (WHO Standard: 100 μg/m^3^).

**Figure 6 ijerph-18-12091-f006:**
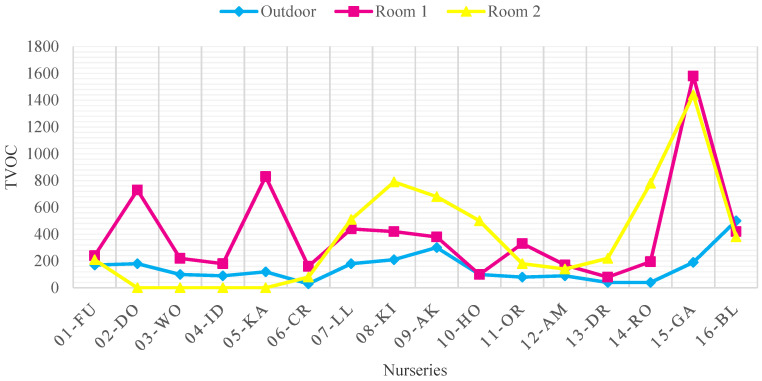
TVOC concentration in each room and outside air before improvement (WHO Standard: 400 μg/m^3^).

**Figure 7 ijerph-18-12091-f007:**
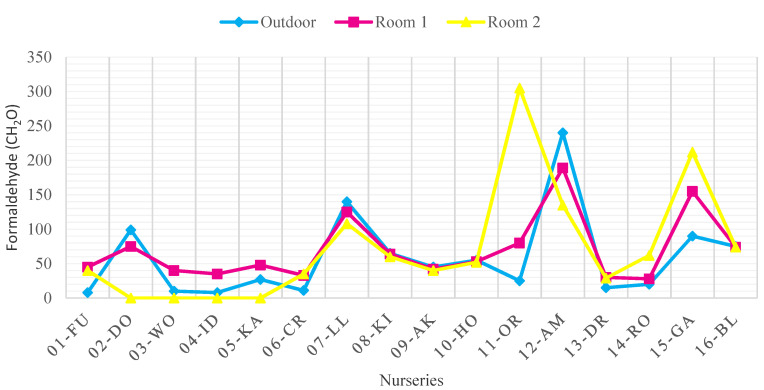
CH_2_O concentration in each room and outside air before improvement (WHO Standard: 100 μg/m^3^).

**Figure 8 ijerph-18-12091-f008:**
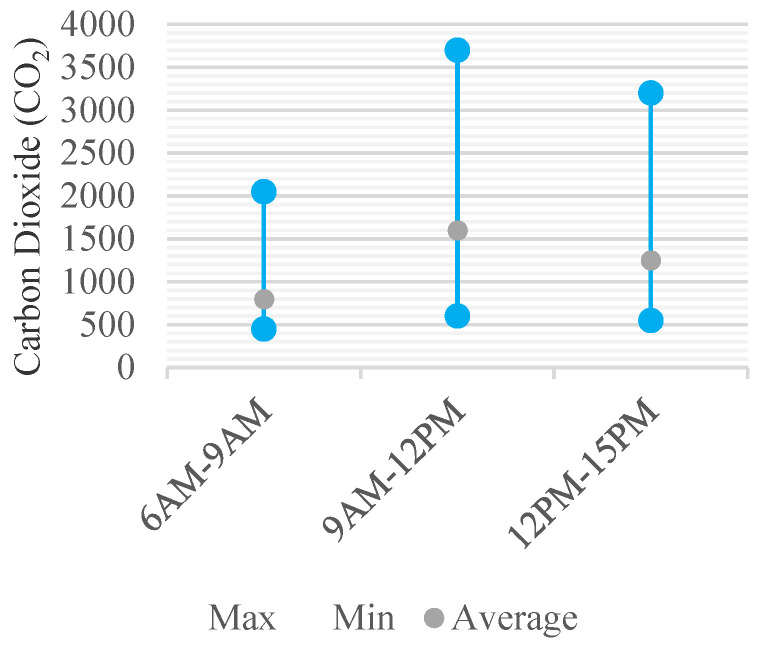
Daily Concentration Change of CO_2_.

**Figure 9 ijerph-18-12091-f009:**
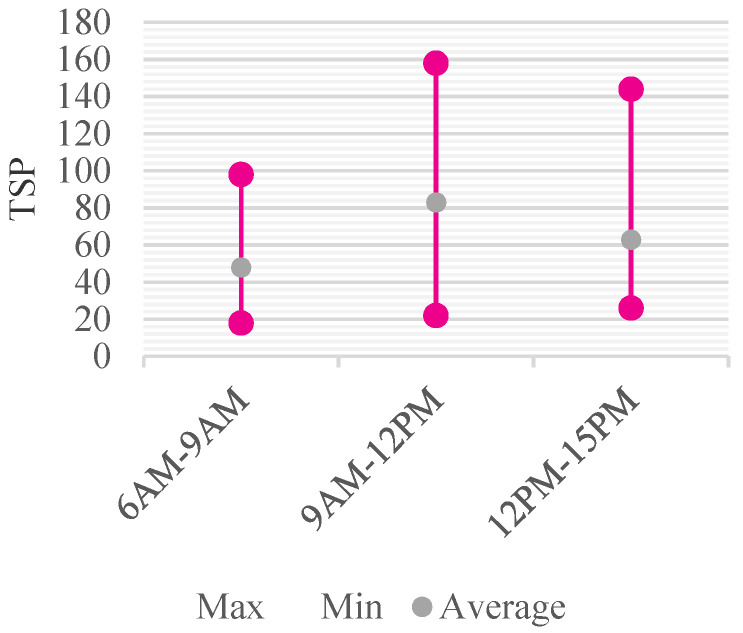
Daily concentration change of TSP.

**Figure 10 ijerph-18-12091-f010:**
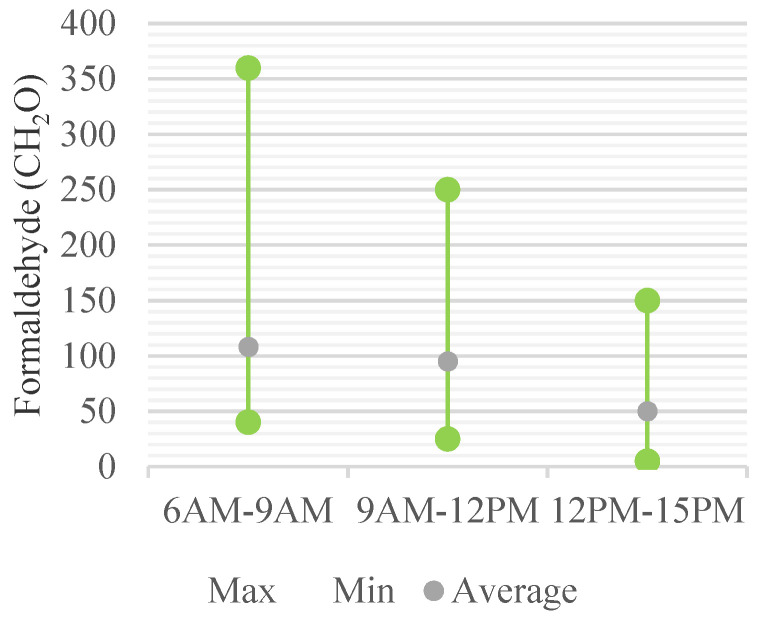
Daily Concentration Change of CH_2_O.

**Figure 11 ijerph-18-12091-f011:**
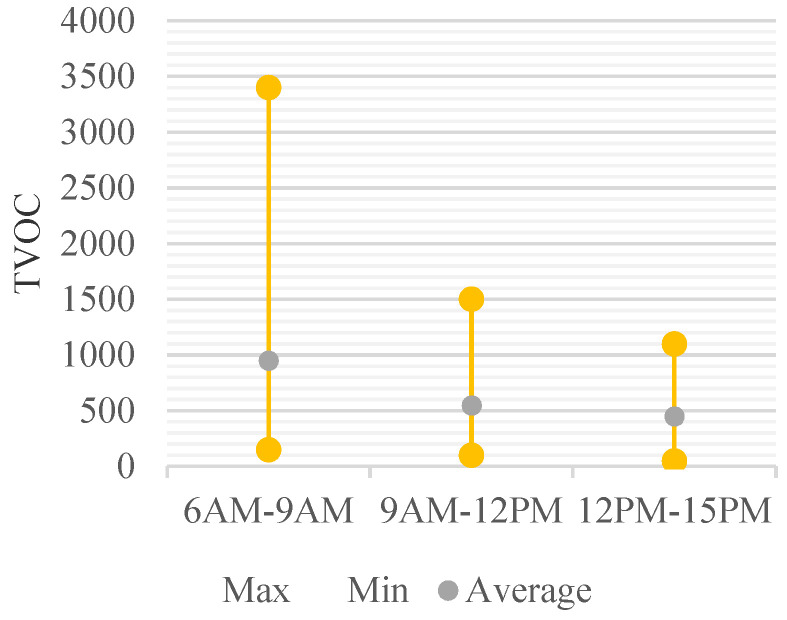
Daily Concentration Change of TVOC.

**Table 1 ijerph-18-12091-t001:** 16 Target Nurseries in United Arab Emirates.

Nursery Name	Facility Basic Information					Management Status				Improvements
	City	Area (m^2^)	Number of Children	Number of Employees	Construction Date	MEP	Natural Ventilation	Daily Cleaning	Annual Sanitization	
01-FU (Figure 2)	Dubai	1120	68	14	2010	Yes	1	3	12	Air Purifier Installation
02-DO	Dubai	708	54	11	2014	Yes	3	3	24	
03-WO	Dubai	810	56	14	2006	Yes	2	1	12	
04-ID	Dubai	1300	104	22	2005	Yes	3	1	1212	
05-KA	Dubai	598	38	10	2010	Yes	3	1	4	
06-CR	Dubai	580	30	8	2012	Yes	4	2	12	
07-LL	Dubai	462	32	6	2016	Yes	3	2	12	
08-KI	Dubai	780	28	9	2008	Yes	4	1	6	
09-AK	Dubai	900	35	10	2006	Yes	2	1	12	
10-HO	Dubai	1200	97	14	2014	Yes	2	1	6	
11-OR (Figure 3)	Sharjah	1310	102	12	2008	Yes	4	4	4	Changing Finish Material
12-AM	Sharjah	588	44	6	2009	Yes	2	1	12	
13-DR	Sharjah	970	83	8	2004	Yes	3	2	12	
14-RO	Sharjah	680	72	7	2008	Yes	3	2	12	New Ventilation Installation
15-GA	Ajman	784	32	6	2014	Yes	4	1	12	
16-BL	Ajman	476	38	4	2016	Yes	2	4	12	

**Table 2 ijerph-18-12091-t002:** Indoor Air Pollutants Measuring Devices and Methods.

Measurement Item	Measuring & Analysis Equipment	Measurement (Sampling) Time
Temperature (°C) Humidity (%)	Electronic Data logger SATO SK-L200TH II	30 Times per Min/Average
Carbon Dioxide (CO_2_)	Indoor Air Quality Meter TSI 7545 IAQ-CALC	30 Times per Min/Average
Formaldehyde (CH_2_O)	Formaldehyde Meter HFX205-100, HPLC	Flow rate: 500 mL/min Collection time: 30 min Collection amount: 15 L 2 times/Average
TVOC	VOC Environmental Meter PCE-VOC 1, Varian-SATURN2200/Shimadzu-QP2010	Flow rate: 100 mL/min Collection time: 30 min Collection amount: 3 L 2 times/Average
TSP	Airmetrics Minivol Portable Air Sampler PAS-201	10 Times per 3 Min/Average

**Table 3 ijerph-18-12091-t003:** Indoor Air Pollutants Measuring Devices and Methods.

Group	Improvement Methods	Target Nurseries	Numbers (Nurseries/Rooms)
AP Group	Installation of Air Purifier	01-FU, 02-DO, 03-WO, 04-ID, 05-KA, 06-CR, 07-LL, 08-KI, 09-AK, 10-HO	10/16
BM Group	Change of Building Materials	11-OR, 12-AM, 13-DR	3/6
NV Group	Installation of New Ventilation	14-RO, 15-GA, 16-BL	3/6

**Table 4 ijerph-18-12091-t004:** Changes in indoor air pollutants before and after improvement with air purifiers.

16 Rooms	CO_2_ (ppm)			TSP (μg/m^3^)			CH_2_O (μg/m^3^)			TVOC (μg/m^3^)		
	Before	After	%	Before	After	%	Before	After	%	Before	After	%
01-FU-1	1962.0	756.0	61.40	88.31	76.46	13.39	46.02	51.66	−12.2	266.42	94.86	64.39
01-FU-2	2114.0	888.0	57.92	98.73	81.12	17.81	41.08	62.68	−52.6	211.36	241.85	−14.41
02-DO-1	858.0	618.6	28.06	94.44	58.98	37.55	71.84	81.02	−12.8	723.93	379.50	47.56
03-WO-1	816.0	557.0	31.50	61.03	58.54	4.05	41.72	63.60	−52.5	229.24	56.55	75.30
04-ID-1	1725.0	1468.0	14.88	84.36	63.81	24.33	35.38	26.04	26.41	174.82	98.14	43.84
05-KA-1	762.0	1477.0	−93.5	61.94	51.55	16.77	47.12	79.74	−69.2	842.63	108.22	87.15
06-CR-1	646.0	694.0	−7.56	78.34	70.45	10.10	31.66	40.86	−29.0	151.42	262.02	−73.02
06-CR-2	714.0	652.0	8.78	57.27	58.76	−2.60	36.32	54.16	−49.1	69.92	30.64	56.15
07-LL-1	564.0	545.0	3.52	84.15	55.20	34.42	121.27	92.48	27.26	489.54	98.28	79.92
07-LL-2	705.0	620.0	12.04	44.17	45.20	−2.78	10.09	63.02	42.74	503.43	62.03	87.67
08-KI-1	1256.0	676.0	46.22	166.60	46.48	72.10	6278	35.15	44.05	467.00	200.53	57.06
08-KI-2	1084.0	1042.0	3.78	182.01	151.80	16.52	62.10	31.68	48.98	781.23	148.98	80.92
09-AK-1	1106.0	431.0	60.92	48.79	35.10	29.50	41.62	62.6	84.94	371.02	50.42	86.38
09-AK-2	1258.0	448.0	64.27	31.02	24.60	20.42	39.98	3.58	91.06	565.01	18.60	96.68
10-HO-1	1366.0	519.0	61.90	103.74	90.50	12.75	57.02	21.45	62.42	108.22	174.55	−61.28
10-HO-2	598.0	978.0	−63.6	85.81	48.88	43.01	55.48	16.74	69.84	199.60	146.55	26.58
Maximum	2114.0	1477.0	64.27	182.01	151.80	72.10	127.27	92.48	91.06	842.63	379.50	96.68
Minimum	564.0	431.0	−93.5	31.02	24.60	−2.78	31.66	3.58	−69.2	69.92	18.60	−73.02
Mean	1097.3	774.65	18.15	85.73	63.60	21.70	56.71	45.64	13.75	384.67	135.73	46.30
Increase	N/A	N/A	3	N/A	N/A	2	N/A	N/A	7	N/A	N/A	3

**Table 5 ijerph-18-12091-t005:** Changes in indoor air pollutants before and after improvement with building material changes.

16 Rooms	CO_2_ (ppm)			TSP (μg/m^3^)			CH_2_O (μg/m^3^)			TVOC (μg/m^3^)		
	Before	After	%	Before	After	%	Before	After	%	Before	After	%
11-OR-1	770.8	711.2	7.78	97.50	71.60	26.56	80.12	479.3	−498	325.46	910.36	−179.8
11-OR-2	512.8	590.8	−15.2	90.81	89.80	1.02	309.41	434.3	−40.4	168.21	715.87	−326.1
12-AM-1	712.8	534.4	24.82	81.24	61.60	24.18	193.12	425.3	−120	148.41	633.58	−326.6
12-AM-1	614.6	1974.5	−221	96.62	66.80	30.98	136.82	231.8	−69.3	98.31	896.41	−812.9
13-DR-1	682.0	593.4	12.90	47.24	40.42	14.48	31.41	112.6	−258	74.31	500.54	−574.1
13-DR-2	816.8	513.2	37.12	55.23	58.43	−5.68	29.54	54.7	−85.1	249.42	412.24	−65.3
Maximum	816.8	1974.5	36.12	97.50	89.80	30.98	309.41	479.3	−40.4	325.46	910.36	−65.3
Minimum	512.8	513.2	−221	47.24	40.42	−5.68	29.54	54.7	−498	74.31	412.24	−812.9
Mean	685.8	820.4	−25.6	78.12	64.76	12.24	130.04	238.7	−178	177.32	678.14	−380.7
Increase	N/A	N/A	2	N/A	N/A	1	N/A	N/A	6	N/A	N/A	6

**Table 6 ijerph-18-12091-t006:** Changes in Indoor Air Pollutants before and after Improvement with New Ventilation Installation.

16 Rooms	CO_2_ (ppm)			TSP (μg/m^3^)			CH_2_O (μg/m^3^)			TVOC (μg/m^3^)		
	Before	After	%	Before	After	%	Before	After	%	Before	After	%
14-RO-1	616.7	436.1	29.3	91.67	25.42	72.27	28.48	7.24	74.57	192.42	189.12	1.72
14-RO-2	636.7	749.3	−17.7	129.17	83.54	35.29	67.98	20.57	69.73	756.32	958.60	−26.72
15-GA-1	2013.3	637.3	68.4	105.56	75.46	28.52	156.86	79.48	49.32	1583.2	875.26	44.52
15-GA-2	2606.7	646.1	75.2	83.33	64.80	22.22	216.08	43.68	79.76	1473.7	690.12	53.12
16-BL-1	586.1	734.8	−25.4	104.27	71.51	31.34	74.54	61.52	17.46	435.46	61.54	86.86
16-BL-2	561.0	991.1	−76.6	184.17	77.44	57.96	73.20	58.70	19.81	389.46	40.94	89.48
Maximum	2606.7	991.1	75.2	184.17	83.54	72.27	216.08	79.48	79.48	1583.2	958.60	89.48
Minimum	561.0	436.1	−76.6	83.33	25.42	22.22	28.48	7.24	17.46	192.42	40.94	−26.72
Mean	1113.1	644.5	17.04	108.46	66.04	36.74	91.94	40.01	53.49	688.48	362.26	52.92
Increase	N/A	N/A	3	N/A	N/A	0	N/A	N/A	0	N/A	N/A	1

## Data Availability

New data were created or analyzed in this study. Data will be shared upon request and consideration of the authors.

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
