# Peer review of "Improving the Indoor Air Quality in Nursery Buildings in United Arab Emirates"

_ijerph, 2021, doi:10.3390/ijerph182212091_

Round 1
Reviewer 1 Report
Dear Authors
My scientific interests are also related to the measures for indoor air quality and its importance to prevent the spread of the COVID-19 pandemic, for instance. Therefore, I am very happy that I could read your very interesting article on this topic.
The manuscript is well-written and has a research and analytic character. The clarity of presented contents helps the reader to enjoy the paper.
In my opinion, in the introduction, the authors should contain a clearly stated aim and hypotheses that the authors should verify in the paper. I think that it is possible to be clearer.
Discussion and conclusions sections should be presented separately. I think that the discussion section should attract other authors on a similar topic.
I really think that this paper will receive a good audience from people who are currently debating the issue of indoor air quality to help to reduce the contagion rhythm of the current pandemic.
If you think that you can add a small section titled for example "Possible applications", where you can join the importance of this paper and COVID literature, please go ahead. You can cite the following papers, with this strategy you will join both pieces of literature.
- Comunian, Silvia, Dario Dongo, Chiara Milani, and Paola Palestini. 2020. Air pollution and covid-19: The role of particulate matter in
the spread and increase of covid-19’s morbidity and mortality. International Journal of Environmental Research and Public Health 17:
4487. - Setti, Leonardo, Fabrizio Passarini, Gianluigi De Gennaro, Pierluigi Barbieri, Maria Grazia Perrone, Massimo Borelli, Jolanda Palmisani,
Alessia Di Gilio, Prisco Piscitelli, and Alessandro Miani. 2020. Airborne transmission route of covid-19: Why 2 meters/6 feet of
inter-personal distance could not be enough. International Journal of Environmental Research and Public Health 17: 2932. - Travaglio, Marco, Yizhou Yu, Rebeka Popovic, Liza Selley, Nuno Santos Leal, and Luis Miguel Martins. 2021. Links between air
pollution and covid-19 in England. Environmental Pollution 268: 115859 - Rodríguez-Caballero, C. V., & Vera-Valdés, J. E. (2021). Air Pollution and Mobility, What Carries COVID-19?. Econometrics, 9(4), 37.
- Shehzad, Khurram, Muddassar Sarfraz, and Syed Ghulam Meran Shah. 2020. The impact of COVID-19 as a necessary evil on air
pollution in India during the lockdown. Environmental Pollution 266: 115080
Author Response
Dear respectful reviewer,
Thank you very much for your sincere review.
Please check the attached revision.

Reviewer 2 Report
This article addresses the very relevant issue of improving indoor air quality in buildings, specifically for children in kindergartens in an area determined by the authors. The article is well structured and very relevant.
Some recommendations we would like to make to the authors are:
1. I think the authors show too many results in the abstract. I recommend to write the result lines in the abstract with less numerical data.
2. Better justify how day-care centres are selected to measure indoor air quality, and identify the change in daily concentration of pollutants.
3. With what level of significance and percentage of error is the research carried out on the sample of the chosen population.
4. Explain in detail the methodological steps with which the fact can be controlled, and that there are no biases with respect to the physical activity of the children combined with the emissions from the building.
5. How effective over other solutions is the installation of a new ventilation system.
6. The authors indicate that renovation with low-emission finishing materials may not improve air quality in a short period of time, but over time it is possible; if they have information on the subject indicate this, if not reinforce it with the publication:
Hormigos-Jimenez, S., Padilla-Marcos, M. Á., Meiss, A., Gonzalez-Lezcano, R. A., & Feijó-Muñoz, J. (2017). Ventilation rate determination method for residential buildings according to TVOC emissions from building materials. Building and Environment, 123, 555-563. https://doi.org/10.1016/j.buildenv.2017.07.032
7. The authors identify the interaction between topics, but do not propose a mechanism to objectively identify the direction of knowledge flow. It would be good to refer to publications on the topic and to improve the state of the art.
Zhang, S., Mumovic, D., Stamp, S., Curran, K., & Cooper, E. (2021, September 1). What do we know about indoor air quality of nurseries? A review of the literature. Building Services Engineering Research and Technology. SAGE Publications Ltd. https://doi.org/10.1177/01436244211009829
Theodosiou, T. G., & Ordoumpozanis, K. T. (2008). Energy, comfort and indoor air quality in nursery and elementary school buildings in the cold climatic zone of Greece. Energy and Buildings, 40(12), 2207-2214. https://doi.org/10.1016/j.enbuild.2008.06.011
Hormigos-Jimenez, S., Padilla-Marcos, M. A., Meiss, A., Gonzalez-Lezcano, R. A., & Feijó-MuÑoz, J. (2018). Experimental validation of the age-of-the-air CFD analysis: A case study. Science and Technology for the Built Environment, 24(9), 994-1003. https://doi.org/10.1080/23744731.2018.1444885.
Valderrama-Ulloa, C., Silva-Castillo, L., Sandoval-Grandi, C., Robles-Calderon, C., & Rouault, F. (2020, January 1). Indoor environmental quality in latin american buildings: A systematic literature review. Sustainability (Switzerland). MDPI. https://doi.org/10.3390/su12020643
8. Mention in the conclusions the research trends in the current scientific community on the topic of the article.
Author Response

(The authors gave the same response as above.)

Round 2
Reviewer 2 Report
The authors have corrected all the comments made and restructured the article as instructed and have improved the quality of the article considerably.
The conclusions and the state of the art have been considerably enriched.